# Post-Mortem Cardiac Magnetic Resonance for the Diagnosis of Hypertrophic Cardiomyopathy

**DOI:** 10.3390/diagnostics10110981

**Published:** 2020-11-21

**Authors:** Giovanni Donato Aquaro, Benedetta Guidi, Federico Biondi, Enrica Chiti, Alessandro Santurro, Matteo Scopetti, Emanuela Turillazzi, Marco Di Paolo

**Affiliations:** 1Fondazione Toscana G. Monasterio, 56124 Pisa, Italy; 2UO Medicina legale, University of Pisa, 56124 Pisa, Italy; benedettaguidi@virgilio.it (B.G.); emanuela.turillazzi@unipi.it (E.T.); marco.dipaolo@unipi.it (M.D.P.); 3Department of Cardiology, University of Trieste, 34100 Trieste, Italy; biondi.federico@yahoo.it; 4Clinical and Translational Science Research Department, University of Pisa, 56124 Pisa, Italy; enricachiti@gmail.com; 5Department of Anatomical, Histological, Forensic and Orthopaedic Sciences, Sapienza University of Rome; 00187 Rome, Italy; alessandro.santurro@uniroma1.it (A.S.); matteo.scopetti@uniroma1.it (M.S.)

**Keywords:** post-mortem cardiac magnetic resonance, hypertrophic cardiomyopathy, sudden death

## Abstract

Background: Post-mortem cardiac magnetic resonance (PMCMR) is an emerging tool supporting forensic medicine for the identification of the causes of cardiac death, such as hypertrophic cardiomyopathy (HCM). We proposed a new method of PMCMR to diagnose HCM despite myocardial rigor mortis. Methods: We performed CMR in 49 HCM patients, 30 non-HCM hypertrophy, and 32 healthy controls. In cine images, rigor mortis was simulated by the analysis of the cardiac phase corresponding to 25% of diastole. Left ventricular mass, mean, and standard deviation (SD) of WT, maximal WT, minimal WT, and their difference were compared for the identification of HCM. These parameters were validated at PMCMR, evaluating eight hearts with HCM, 10 with coronary artery disease, and 10 with non-cardiac death. Results: The SD of WT with a cut-off of > 2.4 had the highest accuracy to identify HCM (AUC 0.95, 95% CI = 0.89–0.98). This was particularly evident in the female population of HCM (AUC=0.998), with 100% specificity (95% CI = 85–100%) and 96% sensitivity (95% CI = 79–99%). Using this parameter, at PMCMR, all of the eight patients with HCM were correctly identified with no false positives. Conclusions: PMCMR allows identification of HCM as the cause of sudden death using the SD of WT > 2.4 as the diagnostic parameter.

## 1. Introduction

Hypertrophic cardiomyopathy is the most common cause of sudden death in young people, with particular predilection for children and young adults [1,2,3]. Sudden death is frequently the first clinical manifestation of this disease, without warning signs or symptoms. Hypertrophic cardiomyopathy is the most frequent cause of sudden death in US competitive athletes [4]. Despite the fact that sudden death is generally associated with vigorous physical exertion, most events are associated with mild exertion or even sedentary activities, including sleep.

HCM is characterized by extreme heterogeneity with regard to phenotypic expression, pathophysiology, and clinical course. It is defined by the presence of increased left ventricular (LV) wall thickness, ranging from moderate (15 mm) to massive (>50 mm) hypertrophy in one or more left ventricular (LV) myocardial segments [1]. The morphologic pattern of hypertrophy is either symmetrical or, more frequently, asymmetrical, and in some cases HCM presents an unusual distribution, as in the apical form.

Histologic findings in patients with HCM reveal hypertrophied myocytes with nuclear pleomorphism, hyperchromasia, and myocardial disarray. However, myocyte disarray is not pathognomonic of HCM, and can also be observed in congenital heart diseases and in normal adult hearts, although it is usually mild and confined to the ventricular free wall septal junctions.

Another common trait of HCM is the myocardial inclusions of areas of fibrosis in the myocardium.

Multiple post-mortem studies document gross macroscopic scarring in HCM patients who died suddenly, and the presence of myocardial interstitial fibrosis or fibrotic replacement at the histopathological level. In addition, small vessel arteries are structurally abnormal, and their lumen and vasodilatory capacity are decreased.

Presently, autopsy and histological examination remain the gold standards for the post-mortem diagnosis of HCM, even if it is likely to be missed by pathologists at necropsy when the ipertrophy is mild and atypical distributed.

Cardiac magnetic resonance (CMR) is considered the gold standard imaging technique for the evaluation of cardiac morphology and function in hypertrophic cardiomyopathy [5]. CMR allows evaluation of cardiac morphology, wall thickness, and ventricular function, permitting diagnosis of HCM with great accuracy and reproducibility. CMR allows myocardial tissue characterization with the identification of myocardial edema and fibrosis, providing prognostic stratification of patients with HCM [6].

In the last few years, post-mortem CMR (PMCMR) has been used as an emerging technique for post-mortem radiological investigation of cardiovascular pathologies. PMCMR of an explanted heart or of the entire body overcomes limitations of spatial resolution, permitting acquisition of sub-millimetric high resolution images [7].

Using high resolution whole heart 3d-SSFP pulse sequence, PMCMR provides information of cardiac morphology with the identification of ventricular abnormalities associated with HCM (such as coronary bridge, crypts, diverticuli, aneurysm, asymmetric or symmetric wall hypertrophy, papillary muscle abnormalities, valvular abnormalities, etc.), and gives quantitative parameters, such as ventricular mass, volumes, and atrial dimensions [7].

In HCM, LV mass could be normal, since a thickened wall is often associated with an opposite thinned one [8]. For this reason, the assessment of the LV mass has shown a low discrimination strength to distinguish HCM from other causes of hypertrophy. Furthermore, the LV mass may be increased in other cardiac conditions, such as cardiac amyloidosis, aortic stenosis, Fabry disease, and systemic hypertension.

In a clinical setting, the diagnosis of HCM is performed using the end-diastolic wall thickness of LV myocardium, but this is obviously not feasible at PMCMR because of rigor mortis, which limits the diagnostic performance of PMCMR for this cardiomyopathy. Analogously to skeletal musculature, when ATP reserves of myocardium are consumed, the heart remains in an early diastolic condition, called rigor mortis. The phase of the cardiac cycle in which the heart becomes paralyzed during rigor mortis is highly variable. Indeed, a recent study by PMCMR in patients with sudden cardiac death showed that the ratio between LV myocardial volume and cavity volume in rigor mortis ranges from 2 to 43, demonstrating that rigor mortis may paralyze the heart in both early and mid-diastole, without any relation to the time interval between death and PMCMR, sex, age, cardiac weight, and body habitus [9].

In the present study, we aimed to define new morphological parameters for the diagnosis of HCM in PMCMR by simulating rigor mortis, as the mid-diastolic cardiac phase of CMR cine images, in living patients with (a) HCM, (b) non-HCM hypertrophy and (c) in healthy controls. Then, these parameters were tested in PMCMR for the identification of HCM among patients with sudden death from other causes.

## 2. Methods

### 2.1. In Vivo Study

We enrolled 50 patients with HCM and known pathogenic mutations of sarcomeric genes, 30 patients with LV hypertrophy of other causes (20 with cardiac amyloidosis and 10 with severe aortic stenosis), and 32 age- and sex-healthy controls.

In vivo CMR was performed using a 1.5T MRI scanner (General Electric Healthcare^®^, Milwaukee, WI, USA) placed at the Fondazione Toscana Gabriele Monasterio, Pisa. A dedicated 16 channel cardiac coil was used.

A set of LV short axis cine images, acquired from the mitral plane valve to the LV ape, were acquired using ECG triggered balanced steady-state free precession (bSSFP) pulse sequence with the following parameters: 30 phases, slice thickness of 8 mm, no gap, eight views per segment, FOV 35–40 cm, phase FOV 1, matrix 224 × 224, reconstruction matrix 256 × 256, 45° flip angle, and a TR/TE near 2 [10].

Analysis of the bSSFP image was made using the MASS^®^ Software Analysis (Leyden, The Netherlands). Briefly, using the software, the end-systolic and the end-diastolic cardiac phases were visually identified as the cardiac phase having, respectively, the lowest and the highest LV volume. Then, a mid-diastolic cardiac phase (25% of diastole) was chosen to simulate the rigor mortis state of the post-mortem heart (Figure 1). In this cardiac phase, we measured LV wall thickness (WT) of all of the conventional 17 myocardial segments using a semiautomatic approach. The LV endocardial and epicardial contour were manually traced with the exclusion of papillary muscles from LV mass. LV mass was measured as the myocardial LV volume multiplied by the myocardial density (1.05 g/cm^3^) [11]. The average LV WT was measured for all of the 17 segments. The following parameters were evaluated: the average of LV WT, the SD of LV WT, maximal WT, the minimal WT, the difference between the maximal and minimal WT (max–minWT), and LV mass.

The presence of secondary features of the HCM phenotype, such as LAD intramyocardial bridge, intramyocardial crypts, and abnormalities of papillary muscles (number and morphology), were also evaluated.

The present study was approved by the local ethics committee (prot. n. 13896, approved on 25 October 2018approved code, approved on day month year) and a written consent was obtained from all of the enrolled patients.

### 2.2. Post-Mortem Cardiac Magnetic Resonance

PMCMR was performed in eight explanted hearts of patients with sudden death and a final histological and genetic diagnosis of HCM. PMCMR was also performed in 10 patients with sudden death and a final diagnosis of coronary artery disease (CAD group) and in 10 patients with sudden death not caused by cardiac disease (non-cardiac death).

PMCMR was carried out using the same 1.5T MRI scanner (General Electric Healthcare^®^, Milwaukee, WI, USA) and 16-channel cardiac coil. A heart rate simulator with a set heart rate of 60 bpm was present. After the acquisition of triplane conventional localizer images, a four-chamber localizer view was acquired using a single-shot SSFP sequence. Then, a whole-heart 3D-Fat Sat prepared SSFP pulse sequence was acquired with following parameters: slice thickness of 0.9 mm, no gap, FOV 22 × 22 cm, phase FOV 1, matrix 256 × 256, reconstruction matrix 512 × 512, flip angle 45°, and a TR/TE ratio approximated to 2. From the 3D images, a set of 2D SSFP LV short axis views from the mitral valve plane to the apex were acquired with the following parameters: slice thickness of 8 mm, no gap, FOV 35-40 cm, phase FOV 1, matrix 224 × 224, reconstruction matrix 256 × 256, 45° flip angle, and a TR/TE near 2. Using the abovementioned MASS^®^ software, the following parameters were calculated: the average of LV WT, the SD of LV WT, maximal WT, the minimal WT, the difference between the maximal and minimal WT (max–minWT), and LV mass (Figure 2). The presence of secondary features of the HCM phenotype, such as LAD intramyocardial bridge, intramyocardial crypts, and abnormalities of papillary muscles (number and morphology), were also evaluated.

### 2.3. Autopsy

According to the guidelines for autopsy investigation of sudden cardiac death, 2017 update from the Association for European Cardiovascular Pathology, gross examination of the heart involved the evaluation of size and morphology, followed by the dissection of the organ through multiple parallel slices from the apex to the mitral valve. The remainder of right and left ventricles in the basal half of the heart was dissected in the direction of the flow of blood for eliciting the gross appearance of cardiac chambers and valves. Mid-cavity free wall thickness of the left and right ventricle and interventricular septum were measured. We measured the WT for each of the 17 conventional myocardial segments of LV. The transverse dimensions of both ventricles and atria were also measured.

Mapped labelled blocks from representative transverse slices of the ventricles were collected to include the apex, the free wall of the left ventricle (anterior, lateral, and posterior), the ventricular septum (anterior and posterior), the free wall of the right ventricle (anterior, lateral, and posterior), and the right ventricular outflow tract.

All samples were stained with haematoxylin and eosin (H&E). When necessary, immunohistochemistry or immunofluorescence for the characterization of inflammatory infiltrates, molecular screening, and electron microscopy were performed.

Conventional autopsy that included macroscopic, histologic and, when appropriate, further laboratory analyses, such as toxicology, chemistry, microbiology, and genetic testing, was performed according to national and international recommendations for forensic autopsy.

The final diagnosis was defined according to the cited guidelines.

### 2.4. Statistical Analysis

Values are presented as the mean ± standard deviation (SD) or as the median (25th–75th percentiles) for variables with normal and non-normal distributions, respectively. Values with non-normal distribution according to the Kolmogorov–Smirnov test were logarithmically transformed for parametric analysis. Qualitative data are expressed as percentages. Categorical variables were compared with the chi-squared test or the Fisher exact test when appropriate. Continuous variables were compared by the ANOVA t-test and analysis of variance or by the Wilcoxon nonparametric test when appropriate. Bonferroni correction was used when needed. Receiver operating characteristic (ROC) curve analysis was used to test the parameter for distinguish between HCM and other patients.

## 3. Results

### 3.1. In Vivo Study

The final population of HCM included 49 patients (one patient excluded for suboptimal image quality for atrial fibrillation), with 24 males and a mean age of 53 ± 12 years in HCM. The group of patients with other causes of LV hypertrophy (non-HCM hypertrophy) included 30 patients, with 18 males and a mean age of 59 ± 12 years. Finally, there were 32 healthy controls, with 15 males and a mean age of 51 ± 15 years.

In HCM groups, 30 (61%) patients had a septal pattern of hypertrophy, eight had septal and apical hypertrophy (16%), two had diffuse hypertrophy (4%), one had inferior-lateral hypertrophy (2%), and eight had apical hypertrophy (16%).

The SD of WT was significantly higher in HCM than in healthy controls (4.1 ± 1.7 mm vs. 1.4 ± 0.4 mm, *p* < 0.001), and in those with non-HCM hypertrophy (4.1 ± 1.7 mm vs. 2.1 ± 0.7 mm, *p* < 0.001). This difference was confirmed both in males (HCM vs. healthy controls: 4.3 ± 1.9 mm vs. 1.5 ± 0.4 mm, *p* < 0.001; HCM vs. non-HCM hypertrophy: 4.3 ± 1.9 mm vs. 2.3 ± 0.7 mm, *p* < 0.001) and females (HCM vs. healthy controls: 3.8 ± 1.3 mm vs. 1.3 ± 0.4 mm, *p* < 0.001; HCM vs. non-HCM hypertrophy: 3.8 ± 1.3 mm vs. 1.6 ± 0.5 mm, *p* < 0.001).

As shown in Table 1, HCM also had a significantly higher maximal WT, minimal WT, mean WT, max–minD and left ventricular mass compared to healthy controls. Compared to non-HCM hypertrophy, HCM had lower minimal WT, max–minD, and LV mass.

Figure 3, Figure 4, Figure 5 and Figure 6 show the ROC curve analysis of LV mass, maximal WT, max–minD, and SD of WT to distinguish between HCM and others.

As evident in the Figures and in Table 2, at ROC curve analysis, the SD of WT with a cut-off of >2.4 had the highest AUC (0.95, 95% CI = 0.89–0.98) to distinguish between HCM and others in the whole population. The effectiveness of this parameter was particularly strong in females (AUC = 0.998, 95% CI = 0.92–1), showing a specificity of 100% (95% CI, 85–100%) and a sensitivity of 96% (95% CI, 79–99%).

Overall, secondary features of the HCM phenotype were found in 28 patients with HCM (57%): LAD intramyocardial bridge in seven patients (14%), intramyocardial crypts in 12 patients (24%), and abnormalities of papillary muscles in 20 patients (40%). Eleven patients had more than one secondary anomaly.

Crypts were also found in one patient of the control group.

### 3.2. PMCMR

PMCMR was performed in eight patients with sudden cardiac death and HCM, 10 with sudden cardiac death and CAD, and 10 with non-cardiac death. Figure 7 shows an example of macroscopic and histological analysis in a case of HCM.

At PMCMR, in patients with a confirmed HCM diagnosis, the maximal WT was 24 ± 6 mm, the SD of WT was 4.6 ± 1.9, and the max–minD was 11 ± 5 mm. In patients with CAD, the maximal WT was 20 ± 4 mm, the SD of WT was 2.1 ± 1.1, and the max–minD was 6 ± 3 mm. In patients with non-cardiac death, the maximal WT was 19 ± 3 mm, the SD of WT was 1.9 ± 0.8, and the max–minD was 5 ± 2 mm.

The maximal WT was not significantly different between HCM and CAD (*p* = 0.10), but it was significantly higher in HCM than in non-cardiac death (*p* = 0.03).

The max–minD was significantly higher in HCM than in CAD (*p* = 0.005) and in non-cardiac death (*p* = 0.0007). The SD of WT was significantly higher in HCM than in CAD (*p* = 0.003) and in non-cardiac death (*p* = 0.0009).

The cut-off of the SD of WT found in the in vivo study (>2.4) was able to detect all of the eight patients with HCM, and none of the patients with CAD and with non-cardiac death. The maximal WT was > 16 mm in seven out of eight patients with HCM, but also in three patients with CAD and in four with non-cardiac death. The max–minD was > 9 mm in six out of eight patients with HCM, in two with CAD, and in none with non-cardiac death.

The SD of WT with a 2.4 threshold had 100% sensitivity (95% CI = 63–100) and 100% specificity (95% CI = 83–100) to identify HCM. The maximal WT had 88% (95% CI = 47–99) sensitivity and 68% (95% CI = 45–86) specificity. Finally, the max–minD had 75% (95% CI = 35–96) sensitivity and 91% (95% CI = 70–98) specificity.

Secondary features of HCM were found in five out of eight (63%) explanted hearts with HCM.

## 4. Discussion

In the present study, we simulated in vivo the rigor mortis of myocardium as the mid-diastolic phase of the cardiac cycle (25% of the diastole) and compared different parameters of LV morphology for the capability of detecting HCM from other causes of hypertrophy or a healthy condition. 

The LV mass, the mean WT, the SD of WT, the maximal and minimal WT, and their difference were calculated.

The main finding was that the SD of WT was the parameter with the highest AUC to identify HCM. This parameter was particularly effective in female patients, for which it demonstrated a 100% specificity and 96% sensitivity to identify HCM.

We tested all of these parameters in explanted hearts of patients dead from cardiac disease (HCM and CAD) and from non-cardiac death. The SD of WT was the unique parameter permitting perfect identification of HCM from CAD and non-cardiac death.

These results are quite relevant because they demonstrated how it is possible to identify the HCM phenotype in rigor mortis, highlighting the potentiality of PMCMR for the diagnosis of the cause of sudden cardiac death.

The morphological phenotype of HCM is heterogeneous. Sixteen different patterns of hypertrophy are described in HCM, but all of them are characterized by asymmetrical hypertrophy [12]. Hypertrophy in HCM is most frequently located in the interventricular septum, but it may also be found in both the septum and apical region, or it may be limited to apical segments or to the inferolateral segments [13]. Yet, in a non-negligible percentage of cases, hypertrophy has a diffuse pattern of distribution, involving the majority of myocardial segments, but in these situations, the WT is also heterogeneous.

The SD of WT is, indeed, a measure of this heterogenous distribution of WT in HCM and therefore, the results of the current study are not surprising.

This new parameter of PMCMR may help to diagnose HCM as the cause of sudden cardiac death. In HCM, histologic findings reveal hypertrophied myocytes with nuclear pleomorphism, hyperchromasia, and myocardial disarray [14]. However, myocyte disarray is not pathognomonic of HCM, and can also be observed in congenital heart diseases and in normal adult hearts, although usually mild and confined to the ventricular free wall septal junctions. Only when assessed quantitatively with > 10% of the heart, disarray is considered diagnostic of HCM. The diagnostic significance of a lower percentage of myocyte disarray has not yet been determined [15].

Another hallmark of HCM is the myocardial inclusions of areas of fibrosis in the myocardium. Furthermore, multiple post-mortem studies document an abnormal structure of the intramural small vessel, with different grades of lumen stenosis [16]. It has been hypothesized that the fibrotic replacement secondary to ischemia and the myocardial scars represents a potentially arrhythmogenic substrate, increasing the patient’s susceptibility to ventricular tachycardia/fibrillation and the risk of sudden death. In the asymmetrical septal variant of HCM, the gross examination may reveal the thickening of the basal anterior septum with subaortic bulging, leading to left ventricular outflow tract obstruction. Septal endocardial friction lesions associated with this finding are also observed. However, the final diagnosis of HCM implies exclusion of abnormal loading conditions, which leads to a secondary ventricular hypertrophy and other diseases that mimic HCM, including cardiac amyloidosis, Fabry disease, glycogen storage disease, and mitochondrial cardiomyopathies. An exercise-induced hypertrophy is also yet to be differentiated [17].

It is worth noting that the SD of WT was particularly able to identify females with HCM. HCM females had a worse prognosis than males [18]. The diagnostic cut-off of ≥15 mm of end-diastolic WT, which is widely accepted as the diagnostic criterion for HCM, is valid for both sexes. However, considering the lower body weight and height, this cut-off could be too high for female patients and be associated with a more advanced stage of disease. In females, the SD of WT could be useful to detect HCM in earlier stages when the maximal end-diastolic WT is <15 mm. Future studies are needed to test this hypothesis.

### Limitations

Some study limitations should be mentioned. First, in the in vivo study, we simulated rigor mortis as the phase of the cardiac cycle corresponding to 25% of the diastole. This may be an over-simplification, because as demonstrated by Bonzon et al [9], rigor mortis may paralyze the heart indifferently in any phase of early and mid-diastole. The WT and derived parameters could change in different diastolic phases. However, the results of the PMCMR study in explanted hearts confirmed the effectiveness of our method to distinguish between HCM and other causes of death, and overcomes this limitation.

Second, the PMCMR study included only HCM and CAD as the causes of sudden death, as well as non-cardiac death. Future studies are needed to evaluate the effectiveness of SD of WT in detecting HCM when compared to other causes of cardiac death.

## 5. Conclusions

Among the different morphological parameters, the SD of WT is the most accurate to identify HCM during rigor mortis. This parameter was tested in a cohort of patients by simulation of rigor mortis, through the analysis of cine images of the mid-diastolic phase of the cardiac cycle, and validated in explanted hearts by PMCMR. The SD of WT > 2.4 is effective in the whole population with HCM, but it is particularly accurate in the female population. Further studies are needed to evaluate its effectiveness in real-life forensic medicine.

## Figures and Tables

**Figure 1 diagnostics-10-00981-f001:**
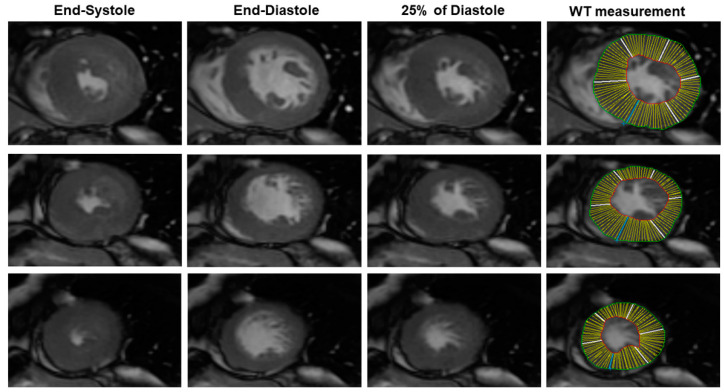
Example of in vivo analysis of cine SSFP images. The cardiac phase corresponding to 25% of the diastole was chosen to simulate rigor mortis. Endocardial and epicardial contours were manually traced excluding papillary muscles. The average myocardial wall thickness (WT) was automatically measured in each of the 17 conventional myocardial segments. The left ventricular (LV) mass, the mean WT, the standard deviation (SD) of WT, the maximal WT, the minimal WT, and the maximal-minimal difference of WT were measured.

**Figure 2 diagnostics-10-00981-f002:**
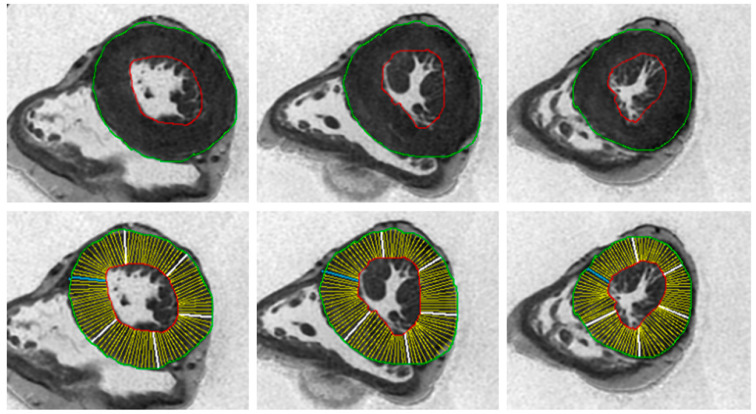
Example of analysis of post-mortem cardiac magnetic resonance images. In these images, the endocardial and epicardial contours were traced, and the myocardial thickness of all of the 17 myocardial segments was measured. The same parameters of Figure 1 were obtained.

**Figure 3 diagnostics-10-00981-f003:**
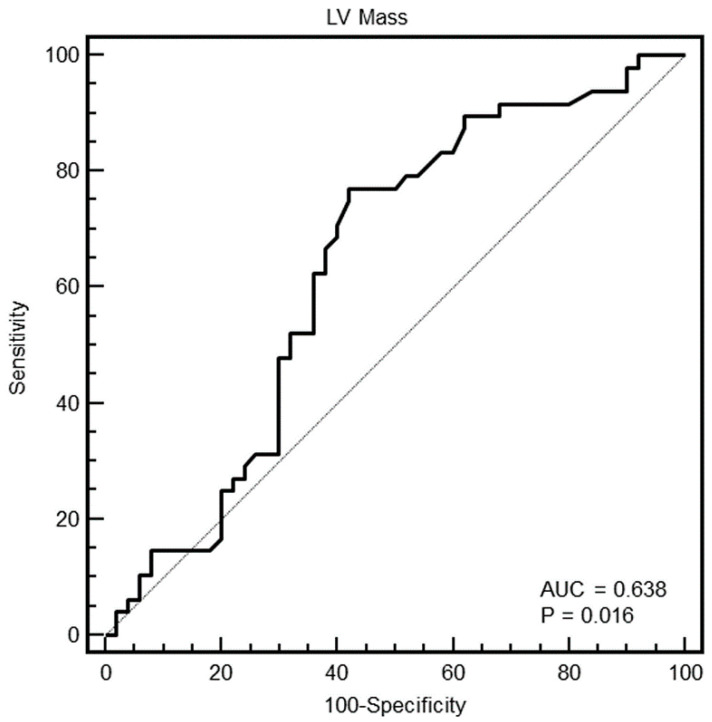
Receiver operating characteristic (ROC) curves of the LV mass for identification of patients with HCM in the whole population. The grey line represents the AUC = 0.5; black line is the ROC curve of LV mass.

**Figure 4 diagnostics-10-00981-f004:**
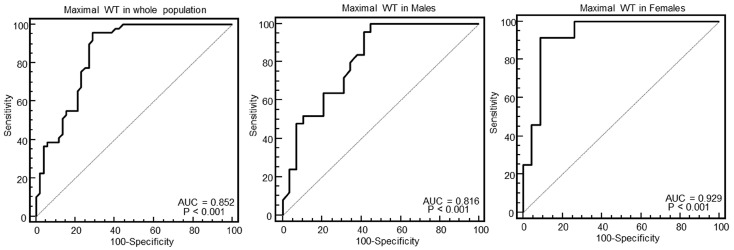
Receiver operating characteristic curves of the maximal wall thickness (WT) for identification of patients with HCM in the whole population, and in males and females.

**Figure 5 diagnostics-10-00981-f005:**
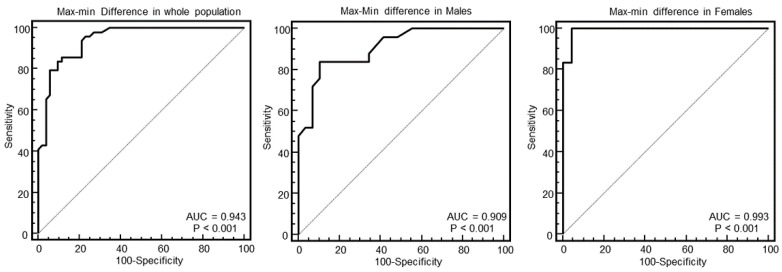
Receiver operating characteristic curves of the difference between maximal and minimal wall thickness (WT) for identification of patients with HCM in the whole population, and in males and females.

**Figure 6 diagnostics-10-00981-f006:**
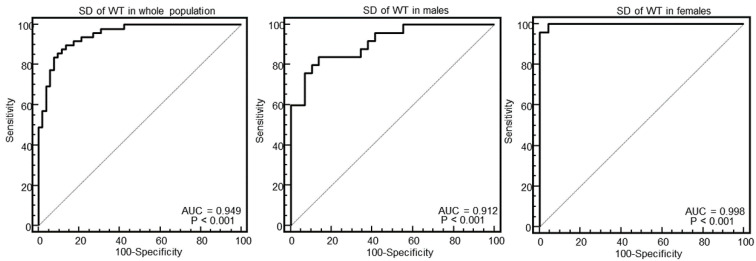
Receiver operating characteristic curves of the standard deviation of wall thickness (WT) for identification of patients with HCM in the whole population, and in males and females.

**Figure 7 diagnostics-10-00981-f007:**
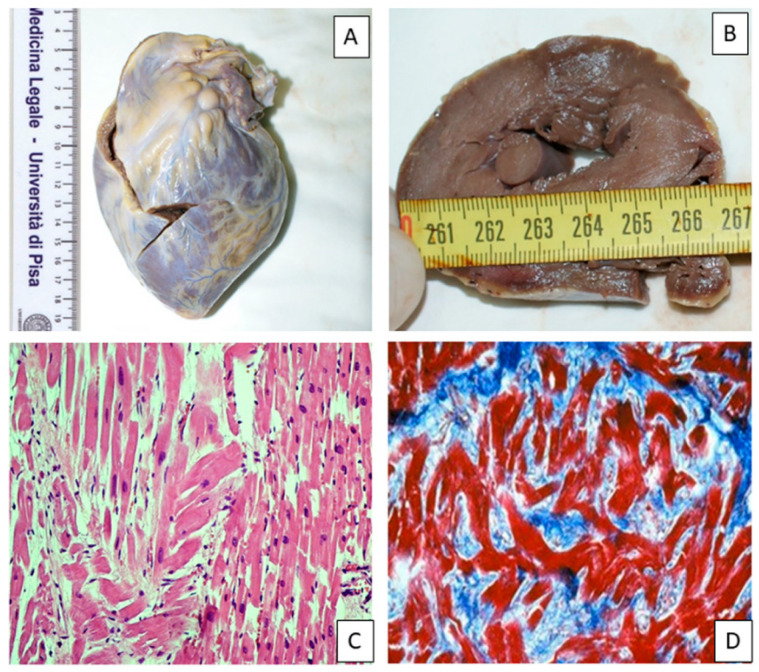
Pathology findings in a case of an HCM patient 22 years old with sudden death. Gross examination 270 gr – 1°. In this section, there is mild left ventricular hypertrophy and mild papillary muscle hypertrophy (**A** and **B**). Histological evaluation of the LV free wall showing cardiomyocyte enlargement and disarray (**C**, hematoxylin and eosin staining, 200×) and interfascicular fibrosis (**D**, Masson’s trichrome staining, 200×).

**Table 1 diagnostics-10-00981-t001:** Comparison of morphometric features between different groups.

	Healthy Controls	*p*-Value ^1^	HCM	*p*-Value ^2^	Non-HCM Hypertrophy
n	32		49		30
Age (years)	48 ± 15	0.3	53 ± 12	<0.001	69 ± 12
Males, n (%)	15 (47%)	0.7	24 (50%)	0.15	18 (60%)
Maximal WT, whole population (mm)	11 ± 2	<0.001	21 ± 5	0.07	19 ± 4
Maximal WT, males (mm)	12 ± 1	<0.001	22 ± 5	0.17	20 ± 4
Maximal WT, females (mm)	11 ± 2	<0.001	20 ± 5	0.07	16 ± 3
Minimal WT, whole population (mm)	6 ± 1	0.004	7 ± 2	<0.001	11 ± 3
Minimal WT, males (mm)	7 ± 1	0.14	8 ± 2	<0.001	12 ± 2
Minimal WT, females (mm)	6 ± 1	0.007	7 ± 2	<0.001	11 ± 4
Mean WT, whole population (mm)	8 ± 1	<0.001	14 ± 3	0.053	15 ± 3
Mean WT, males (mm)	9 ± 1	<0.001	14 ± 3	0.07	16 ± 3
Mean WT, females (mm)	8 ± 1	<0.001	13 ± 3	0.95	13 ± 3
Max–minD, whole population (mm)	5 ± 1	<0.001	14 ± 5	<0.001	8 ± 3
Max–minD, males (mm)	5 ± 2	<0.001	15 ± 6	<0.001	9 ± 3
Max–minD, females (mm)	5 ± 1	<0.001	13 ± 5	<0.001	6 ± 2
LV mass, whole population (g)	102 ± 25	<0.001	160 ± 54	0.03	193 ± 60
LV mass, males (g)	118 ± 17	<0.001	178 ± 52	0.012	221 ± 43
LV mass, females (g)	85 ± 19	<0.001	141 ± 50	0.5	126 ± 34
SD of WT, whole population	1.4 ± 0.4	<0.001	4.1 ± 1.7	<0.001	2.1 ± 0.7
SD of WT, males	1.5 ± 0.4	<0.001	4.3 ± 2.0	<0.001	2.3 ± 0.7
SD of WT, females	1.3 ± 0.4	<0.001	3.8 ± 1.3	<0.001	1.6 ± 0.5

HCM, hypertrophy cardiomyopathy; non-HCM hypertrophy included 20 patients with cardiac amyloidosis and 10 with aortic stenosis. ^1^
*p*-value of healthy controls vs. HCM comparison; ^2^
*p*-value of HCM vs. non-HCM hypertrophy comparison. WT, wall thickness; Max–MinD, difference between the maximal and minimal wall thickness; SD, standard deviation.

**Table 2 diagnostics-10-00981-t002:** Results of ROC curve analysis for distinguishing between HCM and others.

Parameters	Cut-off	Specificity	Sensitivity	AUC	*p*-Value	Standard Error
**Whole population**						
LV mass	>122 g	58 (43−72)	77 (63−88)	0.64 (0.54−0.73)	0.016	0.06
Maximal WT	≥16 mm	71 (57−83)	96 (86−99)	0.85(0.77−0.92)	<0.0001	0.04
Minimal WT	≤11 mm	19 (10−33)	100 (93−100)	0.54 (0.43−0.64)	0.53	0.06
Mean WT	>11 mm	62 (47−75)	92 (80−98)	0.73 (0.63−0.81)	<0.0001	0.05
Max–minD	>9 mm	88 (77−96)	86 (73−54)	0.94 (0.88−0.98)	<0.0001	0.02
SD of WT	>2.4	85 (72−93)	90 (78−97)	0.95 (0.89−0.98)	<0.0001	0.02
**Males**						
LV mass	>130 g	41 (24−61)	84 (64−96)	0.56 (0.42−0.70)	0.44	0.08
Maximal WT	>16 mm	55 (36−74)	100 (86−100)	0.82 (0.69−0.91)	<0.0001	0.06
Minimal WT	≤11 mm	31 (15−51)	100 (86−100)	0.61 (0.47−0.74)	0.14	0.08
Mean WT	>11 mm	52 (33−71)	96 (80−99)	0.65 (0.51−0.74)	0.05	0.08
Max–minD	>10 mm	89 (73−98)	84 (64−96)	0.91 (0.79−0.97)	<0.0001	0.04
SD of WT	>2.6	86 (68−96)	84 (64−96)	0.91 (0.80−0.97)	<0.0001	0.04
**Females**						
LV mass	>122 g	90 (70−99)	65 (43−84)	0.79 (0.64−0.89)	0.0001	0.07
Maximal WT	>16 mm	91 (72−98)	92 (73−99)	0.93 (0.81−0.98)	<0.0001	0.04
Minimal WT	>7 mm	56 (35−77)	67 (44−84)	0.59 (0.44−0.73)	0.28	0.09
Mean WT	>11 mm	83 (61−95)	88 (68−97)	0.86 (0.73−0.95)	<0.0001	0.06
Max–minD	>7 mm	96 (78−99	96 (79−99)	0.993 (0.91−1)	<0.0001	0.008
SD of WT	>2.3	100 (85−100)	96 (79−99)	0.998 (0.92−1)	<0.0001	0.003

WT, wall thickness; Max–MinD, difference between the maximal and minimal wall thickness; SD, standard deviation.

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
