# Peer review of "Post-Mortem Cardiac Magnetic Resonance for the Diagnosis of Hypertrophic Cardiomyopathy"

_diagnostics, 2020, doi:10.3390/diagnostics10110981_

Round 1
Reviewer 1 Report
I read with great interest the manuscript “Post-Mortem Cardiac Magnetic Resonance for the diagnosis of Hypertrophic Cardiomyopathy”. The Authors have proposed a new method of Post-mortem cardiac magnetic resonance (PMCMR) to diagnose HCM despite myocardial rigor mortis.
In cine images, rigor mortis was simulated in vivo by the analysis of the cardiac phase corresponding to the 25% of diastole. Left ventricular mass, mean and standard deviation (SD) of WT, maximal WT, minimal WT and their difference acquired in vivo using CMR were validated at PMCMR.
The SD of WT >2.4 was identified a very good parameter for the identification of HCM as cause of sudden death.
The manuscript is very interesting; however I have some criticisms:
- How many WT measurements were acquired in Autopsy?
- On the contrary respect to method section, the presence of secondary features of HCM phenotype (LAD intramyocardial bridge, intramyocardial crypts and abnormalities of papillary muscles) evaluated by CMR (see method section) were not included in result section. Are secondary features of HCM observed in autopsy?
- Are these PMCMR feature appliable in HCM phenocopies
Author Response
I read with great interest the manuscript “Post-Mortem Cardiac Magnetic Resonance for the diagnosis of Hypertrophic Cardiomyopathy”. The Authors have proposed a new method of Post-mortem cardiac magnetic resonance (PMCMR) to diagnose HCM despite myocardial rigor mortis.
In cine images, rigor mortis was simulated in vivo by the analysis of the cardiac phase corresponding to the 25% of diastole. Left ventricular mass, mean and standard deviation (SD) of WT, maximal WT, minimal WT and their difference acquired in vivo using CMR were validated at PMCMR.
The SD of WT >2.4 was identified a very good parameter for the identification of HCM as cause of sudden death.
The manuscript is very interesting.
Answer: Many thanks for the recognition of the value of our study
Reviewer: How many WT measurements were acquired in Autopsy?
Answer: Thank you for this comment by which we realised that this point was not specified in the method section. We explained it in this new revised manuscript. In the method section (row 156) we included the following statement” We measured the WT for each of the 17 conventional myocardial segments of LV.
Reviewer: On the contrary respect to method section, the presence of secondary features of HCM phenotype (LAD intramyocardial bridge, intramyocardial crypts and abnormalities of papillary muscles) evaluated by CMR (see method section) were not included in result section. Are secondary features of HCM observed in autopsy?
Answer: Many thanks. We missed to insert this other interesting finding. In the “in-vivo” study “ secondary features of HCM phenotype were found 28 patients HCM (57%): LAD intramyocardial bridge in 7(14%), intramyocardial crypts in 12 (24%) and abnormalities of papillary muscles 20 patients (40%). Eleven patients had more than 1 secondary anomaly” (row 222 of results section).
In the PMCMR study, “Secondary features of HCM were found in 5 out of 8 (63%) explanted heart with HCM.” (row 259)
Reviewer: Are these PMCMR feature appliable in HCM phenocopies?
Answer: this is a very interesting question. Unfortunately, in this methodological study we evaluated only patients with “histological and genetical diagnosis of HCM” in order to avoid any confounding factor. Further study are needed to evaluate whether our findings could be applied also to distinguish between HCM and phenocopies.
Reviewer 2 Report
HCM is the most common cause of sudden death, especially among young people. HCM is a very heterogeneous disease. The use of various methods, including PMCMR, is very valuable for understanding the mechanisms and course of the disease. The obtained results can support both forensic medicine and cardiologists and cardiac surgeons. The manuscript was written very clearly, taking into account correctly selected research methods. The manuscript can be accept in present form for the publication in Diagnostics.
Author Response
Answer: Many thanks for the recognition of the value of our study